# Impact of Nutritional Status on Severe Radiation-Induced Mucositis in Oropharyngeal Cancer Patients Undergoing Chemo-Radiotherapy

**DOI:** 10.3390/nu17203301

**Published:** 2025-10-21

**Authors:** África Fernández-Forné, Rocío Fernández-Jiménez, María Dolores Toledo-Serrano, Herminda Jiménez-Rodríguez, Marina Muñoz-Lupiáñez, María Asunción Ruiz-López, José Manuel García-Almeida, Lourdes De la Peña-Fernández, María Isabel Queipo-Ortuño, Jaime Gómez-Millán

**Affiliations:** 1Department of Radiation Oncology, Virgen de la Victoria University Hospital, 29010 Malaga, Spain; mdtsrt@hotmail.com (M.D.T.-S.); hermindajr95@hotmail.com (H.J.-R.); mmlupianez64@gmail.com (M.M.-L.); masunruz@gmail.com (M.A.R.-L.); jaimegomezmillan@gmail.com (J.G.-M.); 2Department of Radiation Oncology, Comprehensive Cancer Unit, Quirónsalud Málaga Hospital, 29004 Malaga, Spain; 3School of Medicine, University of Malaga, 29010 Malaga, Spain; 4Department of Endocrinology and Nutrition, Virgen de la Victoria University Hospital, 29010 Malaga, Spain; rociofernandeznutricion@gmail.com (R.F.-J.); jgarciaalmeida@gmail.com (J.M.G.-A.); 5Málaga Biomedical Research Institute and BIONAND Platform (IBIMA), 29010 Malaga, Spain; maribelqo@gmail.com; 6Department of Endocrinology and Nutrition, QuirónSalud Málaga Hospital, 29004 Malaga, Spain; 7Department of Radiology and Physical Medicine, School of Medicine, University of Málaga, 29010 Malaga, Spain; lpf@uma.es; 8Department of Surgical Specialties, Biochemical and Immunology, School of Medicine, University of Málaga, 29010 Malaga, Spain

**Keywords:** head and neck cancer, radiotherapy, mucositis, nutrition assessment

## Abstract

**Background/Objectives**: Severe radiation-induced mucositis (RIM) is the most distressing acute side effect experienced by oropharyngeal squamous cell carcinoma (OPSCC) patients during chemo-radiotherapy (CRT), with a prevalence between 40 and 68%. RIM severity exhibits a multifactorial etiology that remains unclear. We aimed to analyze nutritional and morphofunctional predictive factors for severe RIM in OPSCC patients undergoing CRT. **Methods**: A prospective cohort study was conducted. Global Leadership Initiative on Malnutrition (GLIM) criteria, bioelectrical impedance vector analysis (BIVA), functional assessment and dosimetric analysis were performed prior to radiotherapy. **Results**: Eighty-two patients were analyzed. Severe RIM affected 46.3% of patients. Severe malnutrition according to GLIM (*p* = 0.011), prolonged Timed Up and Go (TUG) test (*p* = 0.025) and larger PTV54 volume (*p* = 0.049) were independent predictive factors for severe RIM, while higher fat-free mass (FFM) (*p* = 0.006) showed a protective effect. **Conclusions**: These findings highlight the importance of a comprehensive early nutritional assessment for accurately identifying patients at a higher risk of severe RIM.

## 1. Introduction

Oropharyngeal squamous cell carcinoma (OPSCC) is a malignant neoplasm that accounts for 90% of tumors arising in the oropharynx [1]. Changes in sexual behavior, in combination with an escalating prevalence of co-infection with human papillomavirus (HPV), have led to an increased incidence of OPSCC in recent decades in younger populations, particularly among men under 60 years old in high-income countries, which contrasts with the traditionally higher incidence of OPSCC in older populations, associated with tobacco and alcohol consumption [1,2].

Concurrent chemo-radiotherapy (CRT) significantly improves the prognosis of advanced OPSCC and is currently the standard of care [3]. However, despite recent advances [4], this treatment remains strongly associated with a substantial risk of severe radiation-induced mucositis (RIM) [5], in addition to other potential acute and late side effects, including dysgeusia, xerostomia, trismus or fibrosis [6].

RIM is the most common and distressing acute side effect experienced by OPSCC patients undergoing CRT [7]. Several studies have analyzed the prevalence of severe RIM, with results varying between a range of 41–68% [8,9]. The etiopathogenesis of RIM is considered multifactorial, and the predictive factors contributing to its severity remain unclear [10,11]. Sonis et al. described mucositis induced by cytotoxic treatments as a complex and dynamic biological process [12], characterized by a cascade of events that begins with damage to the epithelium lining the mucosa of the digestive tract and the generation of reactive oxygen species (ROS), which in turn activate inflammatory signaling pathways and the release of pro-inflammatory cytokines (TNF-α, IL-1β and IL-6). This amplified response ultimately results in the development of painful mucosal ulceration, which is vulnerable to secondary infections and can lead to treatment interruptions [13], decreasing survival [14] and quality of life (QoL) of patients [15]. Furthermore, all these symptoms appear to be exacerbated in patients who initiate treatment with undernutrition [16], due to its negative consequences in wound healing across different processes such as fibroblast proliferation, angiogenesis, destruction of the extracellular matrix, loss of amino acids (glutamine and arginine), carbohydrate deficiency and a decreased immune response [17,18].

It is well known that locally advanced head and neck cancer (LAHNC) patients are at a high risk of malnutrition, either due to the tumor stage and location or related to symptoms at presentation [19]. A previous analysis of two multicenter prevalence studies [20] reported that, according to the Global Leadership Initiative on Malnutrition (GLIM) criteria, before the start of oncological treatment, 22% of head and neck cancer (HNC) patients were malnourished. Furthermore, during the course of CRT, the proportion of malnourished HNC patients may increase to over 70%, largely attributable to therapy-related side effects [21].

The assessment of nutritional status has evolved in response to the incorporation of different emerging body composition parameters, such as Bioelectrical Impedance Vector Analysis (BIVA) [22,23] and functional measurements, including timed up and go (TUG) test [24], handgrip strength (HGS) [25,26] and ultrasound-based evaluation of the rectus femoris muscle [27]. In the light of these novel parameters, current definitions of malnutrition encompass loss of weight and muscle mass, changes in cell membrane integrity and fluid imbalance [28]. According to the European Society for Clinical Nutrition and Metabolism (ESPEN), malnutrition can be defined as the consequence of nutrient deficiency, which results in altered body composition, including a reduced fat-free mass (FFM) [29].

These parameters provide a more comprehensive view of nutritional status, as supported by the recommendations of the recent expert consensus published by García-Almeida et al. [30], which emphasize the importance of a complete evaluation, based on their prognostic value, beyond the traditional anthropometric assessment (body mass index (BMI) or recent weight loss), which continues to be the most frequently evaluated variable in the literature [9,21,31]. In patients diagnosed with LAHNC, where chronic inflammation, pain, and dysphagia can severely compromise oral intake, the use of morphofunctional assessment tools enables the early detection of metabolic and functional alterations that precede overt clinical malnutrition [32]. Several studies have shown that parameters such as HGS [33] and fat-free mass index (FFMI) [34] are predictive of adverse clinical outcomes, prolonged hospitalization, impaired quality of life, and reduced survival in HNC patients [35]. In addition, malnutrition itself has been identified as a significant risk factor for RIM in patients undergoing radiotherapy or chemotherapy in nasopharynx cancer [16,21,31], and nutritional status has been proposed as a potential biomarker for predicting its occurrence [36,37]. Notably, early nutritional interventions have been shown to improve nutritional status and decrease the incidence of RIM compared with no intervention [38,39,40], emphasizing the critical role of adequate nutrition in this patient population.

Although there is evidence of an association between nutritional status and oral mucositis, to date, no previous studies have evaluated the predictive value of morphofunctional variables in a homogeneous cohort of patients with OPSCC.

We hypothesize that malnutrition and altered morphofunctional parameters prior to CRT are associated with an increased risk of severe radiation-induced mucositis in OPSCC patients, probably related to impaired tissue repair, increased oxidative stress, and an amplified inflammatory response.

In this context, this prospective observational cohort study aims to provide clinically relevant evidence to improve nutritional assessment through the analysis of nutritional and morphofunctional predictive factors for severe RIM in patients with OPSCC undergoing concurrent CRT, with a particular emphasis on the early identification of parameters that may allow for the detection of high-risk patients prior to oncological treatment.

## 2. Materials and Methods

### 2.1. Participants

For this single-center prospective cohort study, all patients diagnosed with OPSCC admitted to the Radiation Oncology department of Virgen de la Victoria University Hospital (Málaga, Spain) between April 2019 and January 2024, who were considered for curative-intent CRT, were evaluated for eligibility. A flow chart diagram of patient’s selection in our study is shown in Figure 1.

The inclusion criteria were an age over 18 years, a good performance status according to the Eastern Cooperative Oncology Group (ECOG) scale ≤ 1, diagnosed with a histologically confirmed OPSCC suitable for treatment with radical radiotherapy and concurrent chemotherapy. OPSCC suitable for radical CRT refers to patients with non-metastatic disease who had not received any prior radiotherapy or chemotherapy for head and neck cancer and were deemed fit for concurrent treatment. Previous incomplete surgery was allowed. Exclusion criteria comprised the presence of distant metastases, palliative intent treatments, previous nasogastric tube (NGT) or percutaneous endoscopic gastrostomy (PEG).

This study was performed in compliance with the ethical principles outlined in the Declaration of Helsinki. Data collection only included medical information required for the investigation after the patient’s authorization in the form of a signed statement of informed consent. The study was approved by the Provincial Research Ethics Committee of Málaga on the 12 March 2019 (Málaga, Spain; reference number 022019/PI16).

### 2.2. Study Design

Eligible patients were notified about the purpose of the study, and written informed consent was requested prior to inclusion. Patient’s decision whether to participate or not in the study did not influence their treatment. Patients were previously assessed by a multidisciplinary committee that agreed on the optimal treatment proposal.

Subsequently, an initial assessment was conducted by the radiation oncologist. Sociodemographic and clinical characteristics such as age, sex, prevalent diseases, toxic habits (tobacco and alcohol), social status, p16 status and other variables were collected during the anamnesis. Tumor staging followed the AJCC 8th edition [41].

Baseline data concerning nutritional status, including classic parameters and a complete morphofunctional assessment, were collected at baseline (1st day of radiation therapy) by a trained nutritionist. A blood test was performed on the same day to obtain nutritional and inflammatory variables. Further details are provided in the following section.

Clinical status data, pain-relieving medication and RIM grade according to the Common Terminology Criteria for Adverse Events version 5.0 (CTCAE V5.0) [42], were recorded weekly during radiation treatment. For the analysis, the maximum RIM toxicity grade recorded for each patient was used.

### 2.3. Nutritional Assessment

#### 2.3.1. Bioimpedance Assessments

Whole-body bioimpedance assessments were performed using a single-frequency bioelectrical impedance (SF-BIA) apparatus (NutrilabTM whole body bioimpedance vector analyzer, Akern^®^, Pontassieve, Italy). This device facilitates the evaluation of body composition parameters by applying an alternating sinusoidal electric current of 800 µA at a frequency of 50 kHz [43]. PhA was calculated and standardized (SPhA) against reference values for sex and age from healthy Italian adults [44,45,46]. BIVA measurements were performed with participants in the supine position and electrodes placed according to standard tetrapolar configurations, following previously described procedures [47]. The technical accuracy of BIVA was evaluated on a daily basis using a precision track. All Rz and Xc values were uniformly measured within a margin of ±1 Ω of the 385 Ohm reference standard [48,49]. PhA was expressed in degrees as arctan (Xc/R) × (180o/π). Individual SPhA value was determined from the sex- and age-matched reference population value by subtracting the reference PhA value from the observed patient PhA value and dividing the result by the respective age- and sex-reference standard deviation (SD).

##### Bioelectrical Impedance Analysis (BIA)

BIA was used to assess various body composition parameters. These included fat mass (FM), fat-free mass (FFM), body cell mass (BCM), total body water (TBW), extracellular water (ECW), Na/K exchange, and hydration percentage (TBW/FFM).

#### 2.3.2. Functional Assesments

For functional assessments, HGS [25,26] and TUG tests [24] were analyzed.

##### Hand Grip Strengh (HGS)

HGS was measured using a JAMAR hand dynamometer (Asimow Engineering Co., Los Angeles, CA, USA) while the patient was seated with the dominant arm’s elbow bent at 90 degrees. Patients were asked to perform three maximal isometric contractions, then the median value was registered.

##### Timed up and Go (TUG)

TUG test involves the measurement of the time, in seconds, taken to stand up from a chair, walk 3 m, turn around, walk back and sit down again [24].

##### Rectus Femoris Quadriceps Evaluation

Rectus femoris quadriceps muscle of the lower extremity was evaluated using ultrasonography [50]. Patients were positioned supine to measure the anteroposterior thickness of the muscle at the lower third of the femur, between the superior pole of the patella and the anterosuperior iliac spine, using a 10–12 MHz probe and a multifrequency linear array (Mindray Z60, Madrid, Spain) [27]. The following parameters were collected: Rectus femoris cross-sectional area (RF-CSA), rectus femoris circumference (RF-CIR), rectus femoris axis (RF-*Y*-axis and RF-*X*-axis), and subcutaneous fat of the leg (L-SAT). Each parameter was measured three times in order to use the mean result for the analysis.

#### 2.3.3. Global Leadership Initiative on Malnutrition (GLIM) Criteria

Malnutrition status was analyzed following the GLIM criteria, that combine both phenotypic and etiologic factors [51]. Phenotypical criteria include unintentional weight loss (5–10% in 6 months or 10–20% over a longer period), BMI below 20 kg/m^2^ for individuals under 70 years and below 22 kg/m^2^ for those over 70 years, and reduced muscle mass, using FFMI calculated from FFM adjusted for the patient’s height. All patients met at least one etiological criterion, as cancer is considered a chronic inflammatory condition. According to these results, patients were classified into three nutritional risk categories: normonutrition, moderate malnutrition or severe malnutrition.

### 2.4. Radiotherapy and Chemotherapy

#### 2.4.1. Radiotherapy

The organs at risk were contoured following the recommendations of Brouwer CL et al. [52], where the oral cavity structure was defined according to the “extended oral cavity” definition, which partially includes the oropharynx area with no inner surface of the lips. The cranial border of this contour is limited by the hard palate mucosa and mucosal reflections near the maxilla. The caudal border includes the base of tongue mucosa and hyoid posteriorly and the mylohyoid muscle and anterior belly of the digastric muscle anteriorly. The anterior border is limited by the inner surface of the mandible and maxilla. The posterior border reaches the soft palate, uvula, and more inferiorly the base of tongue. Finally, the lateral border is determined by the inner surface of the mandible and maxilla.

A dosimetry analysis of the radiation treatment delivered according to the departmental protocol was performed. The volume (cm^3^) covered by the planning tumor volume (PTV) at a prescribed dose of 65 Gy (PTV65) and 54 Gy (PTV54) was analyzed, corresponding to the radical intent area and the high-risk nodal elective area, respectively, performed according to the international practice guideline recommendations [53,54]. The mean dose (Dmean) received by the oral cavity, as well as the percentage of its volume receiving at least 20 Gy, 30 Gy, 40 Gy and 50 Gy (V20, V30, V40, V50, respectively), were also analyzed. Dose constraints for other organs at risk were established according to the recommendations of the RTOG 0225 and RTOG 0615 protocols [55,56].

Patients were irradiated in a 6 MV Linear Accelerator using intensity-modulated radiation therapy (IMRT) technique, following the scheme proposed by the PARSPORT Royal Marsden Hospital trial [4]: 54 Gray (Gy) at 1.8 Gy per fraction was administered in 30 consecutive fractions to elective lymph node levels at discretion of the radiation oncologist. Integrated BOOST until 65 Gy at 2.17 Gy per fraction in 30 consecutive fractions was administered to pathological lymph nodes and primary tumor. The treatment was administered Monday through Friday over the course of 6 weeks.

#### 2.4.2. Chemotherapy

Simultaneously with radiation therapy, cisplatin (CDDP) was administered intravenously as either 100 mg/m^2^ every 3 weeks or 40 mg/m^2^ weekly, based on renal function and patient tolerance following the departmental protocol. All chemotherapy cycles were given concurrently with radiotherapy.

Adjuvant CRT was indicated in selected postoperative cases (uncomplete surgery, positive margins or extracapsular extension) using the same IMRT regimen combined with the same concurrent chemotherapy regimens. The timing for adjuvant CRT started within 6–8 weeks post-surgery, depending on surgical recovery and patient performance status.

### 2.5. Statistical Analysis

When variables did not follow a normal distribution, a logarithmic transformation of the data was used prior to analysis. Continuous variables were compared using Student’s *t*-test and analysis of variance (ANOVA), depending on the existence of 2 or more comparison groups. Paired Student’s *t*-test was used to compare evolution before, during and after treatment and non-parametric U-Mann–Whitney test to compare groups before and after. Chi-squared and Spearman’s rank correlation coefficient for nominal variables and quantitative variables were used to analyze the association between potential predictors. To analyze the correlation between nutritional status variables and RIM, a Pearson’s correlation analysis was performed for continuous variables and a point-biserial correlation for categorical variables. Multivariate analysis was performed using logistic regression to identify independent prognostic factors associated with severe RIM. Receiver Operating Characteristic (ROC) curves were generated to evaluate the diagnostic accuracy of specific predictors of severe RIM. In the statistical analysis, the degree of nutritional status was categorized as normonutrition or moderate malnutrition versus severe malnutrition, according to GLIM criteria. RIM was categorized according to the CTCAE V5.0 scale into mild or moderate RIM (Grades 1–2) versus severe RIM (Grade ≥ 3).

All statistical analyses were performed with SPSS v.20.0 software (SPSS Inc., Chicago, IL, USA), considering a level of *p* < 0.05 as statistically significant.

## 3. Results

### 3.1. Baseline Characteristics

A total of 152 patients diagnosed with OPSCC were admitted at the Radiation Oncology department of Virgen de la Victoria University Hospital between April 2019 and January 2024. Of these, 37 were excluded because they were transferred from other hospitals and were already evaluated by a different nutritionist, 26 did not receive concurrent chemotherapy, 5 received palliative treatment. Two patients were excluded due to incomplete data. A total of 82 OPSCC patients were included for the analysis.

Baseline clinical characteristics and treatment details are presented in Table 1. The mean age was 62 ± 7 years, ranging from 47 to 79 years old. A total of 75.6% of the patients were under 70 years of age. Concerning tumor variables and their treatment, 45.1% of OPSCC patients showed p16 positive status, clinical stage was predominantly T2 (42.7%) and N2 (42.7%). Only 2 patients (2.4%) received adjuvant CRT. Chemotherapy scheme was at discretion of the medical oncologist, and in 80.5% of patients, CDDP 100 mg/m^2^ IV was administered every 21 days. According to GLIM criteria, 31 patients (37.8%) showed moderate to severe malnutrition at baseline. All patients experienced some degree of RIM for a median of 50 ± 27 days, 38 (46.3%) of whom presented severe RIM (CTCAE V5.0 Grade 3), with none of them experiencing Grade 4 or 5 RIM. Socioeconomic characteristics of patients are provided in Appendix A.

### 3.2. Nutritional and Morphofunctional Assessment

The mean BMI at baseline was 26.89 ± 4.63 kg/m^2^. In total, 68.3% of patients were overweight (according to BMI ≥ 25 kg/m^2^) and 3.7% underweight (BMI < 18.5 kg/m^2^). Albumin and prealbumin levels showed a mean of 4 g/dL ± 0.39 and 27.89 mg/dL ± 5.97, respectively, with a mean C-reactive protein (CRP) level of 7.35 mg/dL ± 13.3.

BIVA revealed that patients presented a mean PhA of 5.33° ± 1.02 and mean SPhA of −0.25° ± 2.08. Additionally, the mean FFM and FFMI were 53.78 kg and 18.61 kg/m^2^, respectively, while the mean BCM was 26.89 kg ± 5.66. Functional measurement revealed an overall mean right HGS measured by maximum dynamometry of 35.62 kg ± 9.38, whereas the RF-CSA was 3.86 cm^2^ ± 1.01. Furthermore, the TUG test showed a mean value of 7.7 ± 2.84. These data and the rest of nutritional and morphofunctional assessment parameters are shown in Table 2.

### 3.3. Association of Baseline Nutritional and Morphofunctional Status with Severe RIM

First, to evaluate potential associations between baseline clinical, nutritional, and demographic characteristics with severe RIM, we performed a univariate analysis. Data are presented in Table 3. GLIM status showed a strong association with severe RIM (*p* = 0.001), with 91.7% of severely malnourished patients presenting severe RIM, compared to 38.6% of patients with normonutrition or moderate malnutrition. Severely malnourished patients were found to present RIM for a mean of 18 days longer (*p* = 0.031). It was also observed that this group of patients had a higher need for NGT (*p* = 0.004). Severe RIM patients showed a greater need for opioids and required higher doses for pain management compared to those with lower grades (*p* = 0.004 and *p* = 0.002, respectively). The bioimpedance and functional analysis revealed significant differences, showing that patients with severe RIM presented at baseline a lower FFM (*p* = 0.047), BCM (*p* = 0.04) and RF-CSA (*p* = 0.02) compared to mild or moderate RIM patients, with FFMI showing a tends towards significance (*p* = 0.07). Furthermore, the TUG test at baseline was significantly prolonged in this group of patients (*p* = 0.014). Finally, regarding dosimetric variables, the volume of PTV65 and PTV54, were strongly correlated to severe RIM (*p* = 0.042 and *p* = 0.022, respectively).

### 3.4. Association Between Baseline Characteristics and Severe Malnutrition

An additional analysis was performed to evaluate the association between baseline clinical and treatment characteristics and nutritional status, in order to identify factors related to severe malnutrition at baseline. In the univariate analysis, clinical factors significantly associated with malnutrition were age and T4 stage. Among patients older than 70 years, a total of 14 patients (70%) presented with severe malnutrition, compared with 6 patients (30%) who were 70 years or younger (*p* = 0.025). Regarding tumor stage, 12 patients (71%) with T4 stage presented with severe malnutrition, compared to 5 (29%) with stages T1–T3 (*p* = 0.05). These factors were independently associated with malnutrition in the multivariate analysis with values of *p* = 0.019 (OR = 5.2, 95% CI 1.3–20.9) and *p* = 0.034 (OR = 4.7; 95% CI 1.1–19.8), respectively.

### 3.5. Correlation of GLIM, BIVA Parameters, Functional Status and RIM

We next investigated the potential correlations between nutritional and morphofunctional parameters and RIM. GLIM status, body composition and functional parameters showed positive associations between them. FFM was strongly associated with BCM (*p* = 0.001), RF-CSA (*p* = 0.001) and HGS (*p* = 0.001). Moreover, GLIM was negatively correlated with FFM (*p* = 0.003), BCM (*p =* 0.006), RF-CSA (*p* = 0.02) and HGS (*p* = 0.003). Cronbach’s α test 0.615.

These data and further correlations are shown in Figure 2, and their respective *p*-values are shown in Appendix A.

### 3.6. Performance of Dosimetric Analysis, GLIM Criteria, BIVA Parameters and Functional Status to Predict Severe RIM

Furthermore, we evaluate the ability of dosimetric, nutritional and morphofunctional variables to predict the risk of severe RIM with a logistic regression analysis adjusted for age and sex. In the first model, performed with GLIM criteria to evaluate nutritional status, we found that larger PTV54 volume and the presence of severe malnutrition according to GLIM criteria were associated with an increased risk of severe RIM (*p* = 0.049 and *p* = 0.011, respectively). In the second model, when including additional factors related to body composition and functional status, the association of PTV54 volume was maintained (*p* = 0.031). In addition, it was observed that a prolonged TUG test was associated with an increased risk of severe RIM (*p* = 0.025). In contrast, a higher FFM showed a protective effect against severe RIM (OR = 0.74, 95% CI 0.58–0.89, *p* = 0.006). These data are shown in Table 4. A third model was performed, incorporating BMI, and it showed that BMI was not a significant factor in determining severe RIM. Model 3 is shown in Appendix A.

Finally, ROC curves were generated to examine the ability of these parameters to predict severe RIM in our cohort. PTV54 volume showed an AUC of 0.66 (95% CI: 0.498–0.822), with an optimal cut-off value of 455 cm^3^ For the TUG test, an AUC of 0.763 (95% CI: 0.619–0.906) was obtained, with an optimal cut-off value of 7.44 s. Finally, when analyzing FFM, the AUC was 0.692 (95% CI: 0.534–0.851), with a cut-off value of 51.8 kg. These data are represented together with their p-values in Figure 3. ROC curves stratified by sex, along with sensitivity and specificity data, are presented in Appendix A.

## 4. Discussion

The current study evaluates the relationship between nutritional status, using GLIM criteria, and morphofunctional assessment, including BIVA and functional test, with RIM in OPSCC patients receiving concurrent CRT.

The GLIM criteria are a globally recognized framework for diagnosing malnutrition in adults, developed to standardize and improve the identification and grading of malnutrition in clinical settings [51], potentially offering greater precision than others instruments such as the Subjective Global Assessment (SGA) [57]. According to GLIM criteria, 37.8% of the patients in our cohort presented moderate to severe malnutrition at baseline, confirming the results of other series that evaluate malnutrition before oncological treatment in HNC patients [32]. Regarding the association between malnutrition and CRT toxicity, a recent study published by Wan et al. [58] with 113 patients treated for nasopharyngeal cancer, no significant correlations between patient nutrition status, according to GLIM criteria, and CRT toxicity was found. In our study, GLIM status was strongly associated with severe RIM in terms of frequency (*p* = 0.001) and duration (*p* = 0.031), emerging severe malnutrition according to GLIM criteria as the strongest independent predictor of RIM (OR: 17.71, *p* = 0.009) in our cohort. This discrepancy with Wan’s study might be explained by the fact that most patients were well-nourished (83%), due to the nasopharynx location, which has less impact on the patient’s ability to ingest food than other HNC subsites like OPSCC. To our knowledge, this is the first report of a significant association between RIM and nutritional status according to GLIM criteria in HNC patients.

Regarding body composition measurements, BIVA has emerged as a valuable method to provide information about water, lean and fat mass [59]. In our study, when functional and BIVA parameters were incorporated into the multivariable model, the significance of GLIM-defined malnutrition was lost (*p* = 0.20), showing that a lower FFM was independently associated with RIM severity (OR: 0.86, *p* = 0.013). These results suggest that traditional nutritional assessment, while useful, may not fully capture the complex interplay between body composition and treatment toxicity, and body composition alterations might be more predictive of toxicity than categorical malnutrition definitions. Our results confirm the findings of Wan et al. [58], where FFMI emerged as a strong prognostic factor for treatment-related toxicity in nasopharyngeal cancer patients. In our study, the finding of FFM, instead of FFMI, as a predictive factor for RIM might be due to the characteristics of our cohort, where reduced variability in height, as a result of the small sample size, may limit the power of FFMI normalization, allowing FFM to remain discriminative.

Unlike traditional malnutrition criteria, which rely on weight loss and BMI, BIVA provides detailed information about body composition, allowing a more precise evaluation of fat mass, muscle mass and hydration status, which BMI cannot distinguish. In this manner, BIVA can identify individuals with normal BMI but abnormal fat or muscle mass, improving early detection of malnutrition, obesity or sarcopenia [59]. In our study, BMI was not a significant prognostic factor for RIM, as shown in Appendix A.

As a functional parameter, the TUG test has already been evaluated, showing its predictive value for malnutrition and cancer outcomes in HNC patients [32]. However, regarding the association between TUG test and toxicity, we found only one study, with a heterogeneous elderly cancer cohort [60], including 32 HNC patients, reporting no statistically significant differences in treatment-related toxicities according to TUG test (*p* = 0.11). Our results show that functional impairment, as measured by TUG test, was significantly correlated with RIM as an independent prognostic factor (OR: 1.47, *p* = 0.038), reinforcing the idea that physical performance is intrinsically linked to nutritional status and treatment tolerance. To our knowledge, our study is the first to demonstrate a significant association between TUG test and severe RIM in HNC patients.

These results reveal that BIVA provides a reliable assessment of muscle and cellular integrity, which may better reflect a patient’s resilience to treatment toxicity. Integrating BIVA and functional parameters into routine nutritional screening protocols could refine risk stratification and enable early interventions to mitigate severe RIM. This accurate detection will also assist the implementation of early nutritional interventions for high-risk patients. Such strategies have been shown to allow clinicians to achieve favorable oncological outcomes and reduce treatment-related toxicities [38,39,40]. Future research should explore the role of BIVA as a standardized tool for predicting RIM risk in oncology patients.

Regarding the volume receiving a dose intended to eliminate subclinical disease in HNC patients, it has been shown that this volume can be diminished under certain clinical circumstances, resulting in a consistent reduction in RIM severity [61]. Our results confirm this concept, showing that PTV54 volume is an independent prognostic factor for RIM in the two models performed.

In summary, the study findings support our initial hypothesis that malnutrition and altered morphofunctional parameters prior to CRT are associated with an increased risk of severe RIM in OPSCC patients.

This study has some limitations. It is a single-center study with a relatively small, homogeneous sample, which could introduce selection bias. The study population may not fully reflect the broader OPSCC population, as most patients were male and heavy smokers. Another limitation of this study is the age distribution of the patients included (mean age 62 ± 7 years, ranging from 47 to 79 years old). Although most patients (76%) were younger than 70 years, the study population still reflects a predominantly middle-aged to older group, which may limit the extrapolation of our results to younger cancer patients. Additionally, the specific clinical and demographic characteristics of HPV-positive patients were not analyzed. These patients may have different clinical profiles, which would be interesting to explore in future studies. Therefore, caution is required when extrapolating these findings beyond the study cohort. Future multi-center studies with larger and more diverse cohorts are warranted to validate our findings.

## 5. Conclusions

In patients with OPSCC receiving CRT, the presence of severe malnutrition defined by GLIM criteria, impaired physical function measured by the TUG test, and larger irradiated tumor volumes (PTV54) were independently associated with the development of severe RIM. Conversely, a higher FFM was identified as a protective factor. These findings support the clinical utility of incorporating early morphofunctional nutritional assessment—including body composition and functional testing—into routine care, to improve risk stratification, implement targeted preventive strategies, and potentially enhance treatment tolerance and outcomes.

## Figures and Tables

**Figure 1 nutrients-17-03301-f001:**
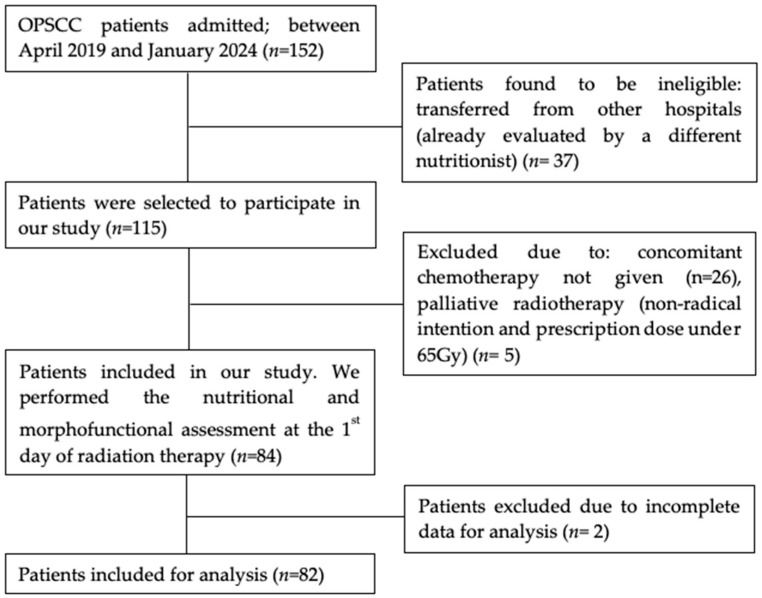
Flow chart diagram. OPSCC: Oropharyngeal squamous cell carcinoma.

**Figure 2 nutrients-17-03301-f002:**
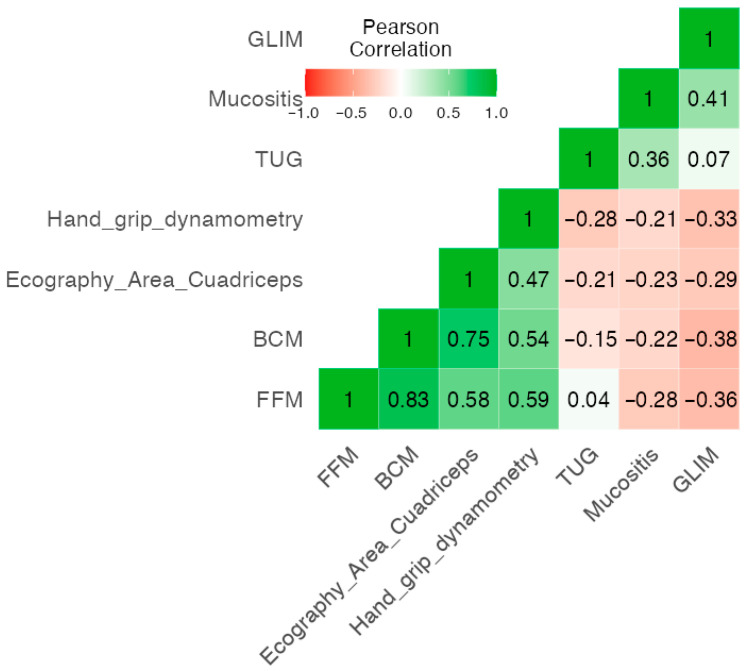
Correlation heatmap of BIVA variables, ultrasound variables, functional test, mucositis grade and nutritional status in head and neck cancer patients undergoing chemo-radiotherapy. FFM: Fat-Free Mass. BCM: Body Cell Mass. Echography area quadriceps (RF-CSA): Rectus femoris cross-sectional area. TUG: Timed Up and Go. GLIM: Global Leadership Initiative on Malnutrition.

**Figure 3 nutrients-17-03301-f003:**
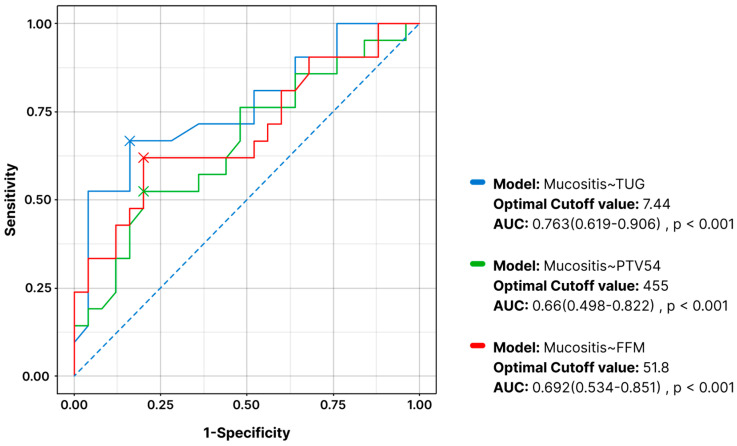
Combined ROC curve analysis of TUG, PTV45 volume and FFM to assess severe radiation-induced mucositis (according to Grade 3 CTCAE V5.0) in oropharyngeal cancer patients undergoing chemo-radiotherapy. ROC: Receiver operating characteristic. TUG: Timed Up and Go. PTV54: Planning Tumor Volume at a prescribed dose of 54 Gy. FFM: Fat-Free Mass. AUC: Area Under the Curve. CTCAE V5.0: Common Terminology Criteria for Adverse Events Version 5.0.

**Table 1 nutrients-17-03301-t001:** General characteristics of patients.

	Variable	Category	N = 82	(%)
Demographic characteristics	Sex	Male	64	(78)
		Female	18	(22)
	Age	<70 years	62	(75.6)
		≥70 years	20	(24.4)
	ECOG	0	43	(52.4)
		1	39	(47.6)
	Smoking	No/stopped	56	(68.3)
		Yes	26	(31.7)
	Cumulative smoking	<10 PYI	14	(17.1)
		≥10 PYI	68	(82.9)
	Alcohol	No/stopped	40	(48.7)
		Yes	42	(51.2)
Tumor characteristics	HPV status (p16)	No/Unknown	45	(54.9)
		Yes	37	(45.1)
	Primary stage	Tx	1	(1.2)
		T1	10	(12.2)
		T2	35	(42.7)
		T3	19	(23.2)
		T4	17	(20.7)
	Nodal stage	N0	9	(11)
		N1	34	(41.5)
		N2	35	(42.7)
		N3	4	(4.9)
	TNM stage (AJCC 8°Ed)	I	14	(17.1)
		II	17	(20.7)
		III	23	(28)
		IV	28	(34.1)
	Differentiation	I	4	(4.9)
		II	50	(61)
		III	28	(34.1)
Treatment characteristics	Treatment option	CRT	80	(97.6)
		Adjuvant CRT	2	(2.4)
	Systemic therapy	Every 3 weeks	66	(80.5)
		Every week	16	(19.5)
Nutritional status and radiation therapy toxicity	GLIM status	Normonutrition	51	(62.2%)
		Moderate malnutrition	19	(23.2%)
		Severe malnutrition	12	(14.6%)
	RIM (CTCAE V5.0)	Grade 0	0	(0%)
		Grade 1	11	(13.4%)
		Grade 2	33	(40.2%)
		Grade 3	38	(46.3%)

ECOG: Eastern Cooperative Oncology Group. PYI: Pack Year Index. HPV: Human Papillomavirus. p16 was used as surrogate marker for HPV status. TNM: Tumor–Node–Metastasis. AJCC: American Joint Committee on Cancer, 8th Edition. CRT: Concurrent chemo-radiotherapy. Adjuvant CRT: Adjuvant chemo-radiotherapy. GLIM: Global Leadership Initiative on Malnutrition. RIM: Radiation-Induced Mucositis. CTCAE V5.0: Common Terminology Criteria for Adverse Events Version 5.0.

**Table 2 nutrients-17-03301-t002:** Nutritional and morphofunctional assessment parameters.

Nutritional Assessment	Variable	Mean (N = 82)	SD
Classic parameters	BMI (kg/m^2^)	26.89	4.63
	Glucose (mg/dL)	106.6	29.68
	Albumin (g/dL)	4	0.39
	Prealbumin (mg/dL)	27.89	5.97
	CRP (mg/L)	7.35	13.3
	C-peptide (ng/mL)	1.95	0.95
Bioimpedance analysis	PhA (°)	5.33	1.02
	SPhA	−0.25	2.08
	FFM (kg)	53.78	8.04
	FFMI (kg/m^2^)	18.61	2.44
	BCM (kg)	26.89	5.66
Functional measurement	HGS (kg)	35.62	9.38
	RF-CSA (cm^2^)	3.86	1.01
	TUG (seconds)	7.7	2.84

SD: Standard Deviation. BMI: Body Mass Index. CRP: C-reactive protein. PhA: Phase Angle. SPhA: Specific Phase Angle. FFM: Fat-Free Mass. FFMI: Fat-Free Mass Index. BCM: Body Cell Mass. HGS: Handgrip Strength. RF-CSA: Rectus femoris cross-sectional area. TUG: Timed Up and Go.

**Table 3 nutrients-17-03301-t003:** Univariate analysis.

Variable	Category	Mild-Moderate RIM N = 44 (54%)	Severe RIM N = 38 (46%)	*p* Value
Active smoker	No	33 (58.9)	23 (41.1)	0.16
	Yes	11 (42.3)	15 (57.7)	
Cumulative smoking (PYI)	<10	10 (71.4)	4 (28.6)	0.14
	≥10	34 (50)	34 (50)	
GLIM status	Normonutrition/ moderate malnutrition	43 (61.4)	27 (38.6)	**0.001**
	Severe malnutrition	1 (8.3)	11 (91.7)	
PTV65 (cm^3^)		143.9 ± 77.2	181.6 ± 87.4	**0.042**
PTV54 (cm^3^)		388.2 ± 112.4	448.3 ± 119.3	**0.022**
FFM (kg)		55.9 ± 6.2	51.3 ± 9.2	**0.047**
FFMI (kg/m^2^)		19.2 ± 1.4	17.9 ± 3.1	0.07
BCM (kg)		28 ± 4.4	4.8 ± 6.6	**0.04**
RF-CSA (cm^2^)		4.08 ± 0.94	3.45 ± 1.07	**0.02**
TUG (s)		6.73 ± 1.63	8.73 ± 3.45	**0.014**

Groups were divided according to RIM (radiation-induced mucositis) status following CTCAE V5.0 (Common Terminology Criteria for Adverse Events Version 5.0) grading. Chi-squared test (or Fisher’s exact test) was used for categorical variables, expressed as absolute numbers and percentages (*p* < 0.05). For continuous variables, expressed as mean ± SD (standard deviations), Student’s *t*-test (or Mann–Whitney test) (*p* < 0.05) was used. Absolute numbers of patients were used for calculating the *p*-values of categorical variables. **Bold** indicates a significant difference between groups. PYI: Pack Year Index. GLIM: Global Leadership Initiative on Malnutrition. PTV65 and PTV54: Planning Tumor Volume at a prescribed dose of 65 Gy and 54 Gy, respectively. FFM: Fat-Free Mass. FFMI: Fat-Free Mass Index. BCM: Body Cell Mass. RF-CSA: Rectus femoris cross-sectional area. TUG: Timed Up and Go.

**Table 4 nutrients-17-03301-t004:** Multivariate analysis adjusted for age and sex. Models 1 and 2.

**Model 1**			
**Variable**	***p* Value**	**OR**	**CI 95%**
Age	0.704	1.01	0.94–1.09
Sex	0.717	1.26	0.35–4.47
PTV54	0.049	1.01	1.00–1.01
GLIM	0.011	16.79	2.76–328
**Model 2**			
**Variable**	***p* Value**	**OR**	**CI 95%**
Age	0.54	1.04	0.91–1.21
Sex	0.34	0.28	0.01–3.42
PTV54	0.031	1.01	1.00–1.03
FFM	0.006	0.74	0.58–0.89
TUG	0.025	1.83	1.20–3.44

Model 1: Predictive factors for severe oral radiation induced mucositis. Model 2: Predictive additional factors for severe oral radiation-induced mucositis (body composition and functional status). OR: Odds Ratio. CI: Confidence Interval. PTV54: Planning Tumor Volume at a prescribed dose of 54 Gy. GLIM: Global Leadership Initiative on Malnutrition. FFM: Fat-Free Mass. TUG: Timed Up and Go.

## Data Availability

The original contributions presented in the study are included in the article/Appendix A, further inquiries can be directed to the corresponding author.

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
