# Peer review of "Impact of Nutritional Status on Severe Radiation-Induced Mucositis in Oropharyngeal Cancer Patients Undergoing Chemo-Radiotherapy"

_nutrients, 2025, doi:10.3390/nu17203301_

Round 1

Reviewer 1 Report

Comments and Suggestions for Authors

The article addresses a clinically relevant issue with a solid prospective design and appropriate statistical methods, integrating GLIM, BIVA, functional and dosimetric variables. However, the text is too verbose in methods and discussion, with unnecessary repetition of data and over-detailed technical descriptions (e.g. electrode placement). Tables are dense and not reader-friendly, especially Table 1 which includes excessive socio-demographic detail of limited clinical interest, and Table 3 which reports too many non-significant variables that could be moved to supplementary material. Figures, particularly the heatmap and ROC curves, lack clarity and require more intuitive presentation. The discussion reiterates results instead of focusing on clinical translation and underplays key limitations such as single-center design, small cohort, and lack of validation. Patient-reported outcomes and potential impact of nutritional interventions are missing. 

Here my line-specific suggestions:

  • Line 137–157: shorten technical details of BIVA electrode placement, refer to prior studies instead.
  • Table 1: move non-essential socio-demographic categories (marital, laboral status) to supplementary.
  • Table 3: retain only significant predictors, relegate the rest.
  • Line 366–377: cut repetition of background already provided in introduction.
  • Line 439–441: expand on external validity and sample bias.
  • Line 442–452 (Conclusions): make it concise, emphasizing malnutrition, FFM, TUG, and PTV54 as predictors with clinical applicability.
Comments on the Quality of English Language

Minor editing is required 

Reviewer 2 Report

Comments and Suggestions for Authors

Review of the manuscript for journal in MDPI

Journal: Nutrients
Manuscript ID: nutrients-3912900
Type of manuscript: Article

The manuscript entitled „Impact of Nutritional Status on Severe Radiation-Induced Mucositis in Oropharyngeal Cancer Patients Undergoing Chemo-Radiotherapy” presents the analysis of nutritional and morphofunctional parameters of patients with oropharyngeal squamous cell carcinoma who are undergoing chemo-radiotherapy and may experience mucositis after radiotherapy.

After radiotherapy, patients without previous omega-3 supplementation often experience burns and inflammation. This is a significant problem that requires resolution. This work contributes to progress in preventing severe radiation-induced inflammation in cancer patients.

Although the work provides valuable information, it is not without inaccuracies.

My comments:

1) Thanks to online access, scientific papers are read not only by specialists in the field but also by patients and their families seeking help. Therefore, all abbreviations should be explained under the tables. I believe the abbreviation QT-RT in Table 1 is poorly described, and ECOG is not explained (also under Table 3), as are others. Please check.

2) There is a lack of consistency between the title, purpose, and conclusions in the work. This should be analyzed (whether we are describing the impact of nutritional status or analyzing predictive factors (abstract)). Does the paper generate clinically useful evidence to improve the assessment of nutritional status or independent predictors of the development of severe RIM? (text).

The paper should be read and any shortcomings corrected.

3) I suggest including the flowchart Figure S1 as Figure 1 in the Materials and Methods chapter in the paper, in the appropriate place, i.e., 2.1.

4) I also suggest using the knowledge recorded in the discussion to modify the conclusions to make them more readable.

I hope my suggestions will improve the readability of the paper.

Reviewer 3 Report

Comments and Suggestions for Authors

The manuscript nutrients-3912900 has an interesting purpose: to evaluate the impact of nutritional status on radiation-induced mucositis in patients with OPSCC undergoing CRT. The manuscript contains 56 references, 25 of which were published in the last 5 years, and 5 self-citations.  

Introduction

Please expand this section with more data regarding the relationship between the importance of adequate nutrition in oral cancer prognosis and the adverse reactions of CRT. Some considerations regarding the age-related incidence of oral cancer are welcome. Based on the existing literature data, please provide some hypotheses. 

Materials and Methods

The following comments are available below:

1. For better understanding, the authors are invited to organize this section as follows:

A. Participants: inclusion/exclusion criteria, well-defined and ethical aspects

Lines 117-118: Please clarify what represents "OPSCC suitable for treatment with radical radiotherapy and concomitant chemotherapy." Were they at the beginning of the CRT treatment or had they previously received a similar therapy?

B. Study design - a brief presentation, mentioning in a general manner the parameters measured, the treatment applied, and the study period for each participant. Were all evaluations performed only at the beginning?  When was the mucositis severity evaluated?

C. Baseline data of the cohort: description and evaluation mode.

D. Nutritional assessments (the current subsection 2.2.): for each category and type of nutritional parameter, there should be a sub-subsection, appropriately entitled, with a principle of each method, description, and supporting references;

E. CRT therapy (subsection 2.3.) should be presented in 2 different sub-subsections:

2.3.1. Radiation therapy

  • preliminary details (lines 194-202)
  • protochol 
  • doses
  • duration

2.3.2. Similar requests are available for Chemotherapy.

Systemic therapy may involve chemotherapy; please explain when it is administered at 3 weeks and when it is necessary every week.

Please provide clearer information about the Adjuvant QT-RT, including the timing of its application and more detailed data (patient's status, dose, duration, and frequency).

F. When the mucositis status was evaluated? Was the period the same for each participant or was it different?

2. The statistical analysis is detailed enough in the current manuscript. However, other correlations are suggested to support the findings.

Results

Table 3. Please show the values being compared for calculating the p-value (the number of patients or percentages) in the baseline data.  The mean age is 62±7; please provide the minimum and maximum ages of the included patients. 

Please correlate the baseline data and the treatment particulars with nutritional status and identify the most significant factors contributing to moderate and severe malnutrition.

The same correlations can be performed considering the mucositis status. 

Discussions

Please discuss the requested correlations and support them with the literature data.

Please show if the study findings support the hypotheses mentioned in the Introduction.  

Please include the age range (62 ± 7) of the patients included in this study in the limitations section, as it suggests that they were predominantly elderly patients (classified into two groups: <70 and> 70 years).

Round 2

Reviewer 3 Report

Comments and Suggestions for Authors

The authors responded to all concerns from the previous review report. There are no more comments available.